# Paediatric Type1 Diabetes Management and Mothers’ Emotional Intelligence Interactions

**DOI:** 10.3390/ijerph18063117

**Published:** 2021-03-18

**Authors:** Jolanta Žilinskienė, Linas Šumskas, Dalia Antinienė

**Affiliations:** 1Department of Health Psychology, Faculty of Public Health, Lithuanian University of Health Sciences, A. Mickevičiaus Street 9, LT-44307 Kaunas, Lithuania; dalia.antiniene@lsmuni.lt; 2Department of Preventive Medicine and Health Research Institute, Faculty of Public Health, Lithuanian University of Health Sciences, A. Mickevičiaus Street 9, LT-44307 Kaunas, Lithuania; linas.sumskas@lsmuni.lt

**Keywords:** type 1 diabetes, emotions, emotional intelligence, health behaviour, emotions, diabetes management

## Abstract

The functioning of the parents’ emotional sphere is very important to a child’s mental and physical health. This study focused on investigating the association between mothers’ emotional intelligence (EI) and paediatric type I diabetes (T1DM) disease management in their children. We hypothesized that mothers’ EI is associated with T1DM outcomes. Mothers of children with T1DM aged 6–12 years were surveyed. One hundred and thirty-four mothers, the main caregivers of their diabetic children, provided measures of EI and completed a demographic questionnaire. The primary indicator of diabetes management was haemoglobin A1c (HbA1c; the main form of glycosylated haemoglobin). EI scales and subscales were associated with glycaemic management indices. Logistic regression analysis was applied for the assessment of the association between parents’ EI and their paediatric with T1DM disease management. The analysis demonstrated a statistically significant correlation between T1DM management and mothers’ ability to understand and control own emotions, to transform their own negative emotions into positive and to control own negative emotions. Mothers’ EI scales and subscales of understanding and regulating their own emotions, subscales of transforming their own negative emotions into positive ones and controlling their own negative emotions were statistically reliable predictors of glycaemic control in children with T1DM.

## 1. Introduction

Type 1 diabetes mellitus (T1DM) is a lifelong metabolic disorder, typically arising in childhood and adolescence. Every year, approximately 80,000 children develop type 1 diabetes (T1DM) worldwide [1]. According to the Lithuanian Institute of Hygiene, in 2018, 818 children and teenagers with T1DM were living in Lithuania (79 children aged 0–6 years, 506 children aged 7–14 years old and 233 children aged 15–17 years) (last viewed 18 January 2021, link: https://stat.hi.lt/default.aspx?report_id=168). Every year in Lithuania, more than 100 children are diagnosed with T1DM diabetes. Diabetes management requires significant involvement of medical/health and financial resources as well as proper parenting involvement. Blood glucose monitoring, insulin administration and the treatment of low blood glucose are essential for children with T1DM during the day, although the need for assistance varies across age groups [2]. Current clinical care guidelines recommend that parents assume the majority of daily diabetes self-care during [3]. Insufficient T1DM management can lead to serious short- and long-term complications and premature death [3]. A diagnosis of T1DM is life-changing not only for children and teenagers, but especially for parents. T1DM impacts and disrupts not only the affected child but the whole family. When T1DM is diagnosed, families have to make multiple life changes simultaneously and in a very short time frame.

Compulsory transformations produce stress in families and may contribute to individual members’ feelings of loss of normalcy and imbalance during this time [4]. The burdensome T1DM management regimen places high demands on parents, children and youth and generates difficulties in terms of psychosocial adjustment [5]. Everyday disease management demands cause many intense emotions and feelings for both the child and the parents. The diagnosis of diabetes affects the whole family and parental reactions of stress, anxiety and depression are common after the diagnosis and impair parents’ psychological functioning [6].

Many research articles have demonstrated medium or strong associations between diabetes-specific distress and depression. A higher frequency of stress and difficulties in parents of young children are associated with higher parental depressive symptoms and fear [7]. Disease-specific demands are associated with elevated symptoms of depression in mothers [8,9,10,11]. Maternal diabetes-related stress and symptoms of anxiety and depression are also correlated [12].

The parents of children and adolescents with T1DM are more anxious and perceive less family cohesion than the parents of healthy children and adolescents [13]. Parents with depressive symptoms have also shown a smaller reduction in daily T1DM-specific distress from baseline to six months of follow-up and reported higher distress across all assessment points, with higher long-term T1DM-specific distress compared to parents without depressive symptoms [14].

Parents’ psychological health is associated with T1DM management adherence behaviours on the part of both the parent and the child, as well as health outcomes [15,16]. Fear of hypoglycaemia in the parents of young children with T1DM is correlated with higher blood glucose levels in young children. Parents’ fear of hypoglycaemia provokes parent stress, which in turn leads to paediatric metabolic control [17]. Maternal depressive symptoms have been correlated with more conflict and less parental monitoring. Conflicts are associated with poorer adherence and glycaemic control. Poorer youth adherence is associated with more conflict, which in turn relates to maternal depressive symptoms [11]. Metabolic control from caregivers is associated with emotional difficulties in adolescents. Adolescent emotional well-being is associated with adherence and/or metabolic control [18].

As many studies have shown, T1DM is an emotional challenge for children and their parents, and there is a connection between parents’ emotional well-being and the child’s T1DM management and emotional well-being. It is very important to understand how parents’ abilities or personality traits can directly and indirectly help cope with the emotional strain of T1DM challenges.

The concept of emotional intelligence (EI) may offer a better understanding of parents’ personal resources, facilitating the management of such challenges. EI is a set of abilities including ‘the ability to monitor one’s own and others’ feeling and emotions, to discriminate among them and to use this information to guide one’s thinking and actions’ [19]. EI is the ability to perceive, assimilate, understand and regulate emotions [19] or the ability to recognise one’s own emotions and those of others, to motivate oneself and to manage emotions both in relation to oneself and others [20]. EI is understood as a delicate intersection of the emotional and intellectual spheres. Research on EI has developed main two trends that reflect different understandings about EI. In the classical direction, EI is seen as a set of abilities leading to a new form of intelligence operating on emotional information. According to this model, EI comprises four abilities: perception (accurately perceiving emotions in oneself and others), facilitation (using emotions to facilitate task performance), understanding (understanding relationships between emotions and situations over time), and regulation (regulating one’s own and others’ emotions) of emotions. New research speaks about EI as a particular dimension of personality that defines traits of personality’s emotional sphere. Trait EI models conceptualise EI as a variety of non-cognitive traits, behavioural dispositions and motivational variables that are linked to successful coping with environmental demands [21]. Depending on the nature of EI, scientific approaches differ [22], but at the same time there have been attempts to combine them [23]. This interaction is particularly reflected through the impact of EI on mental health. EI is positively associated with psychological and physiological health [24,25]. Researchers have identified the importance of individual characteristics that prevent psychological distress, and promote psychological well-being, among which trait EI has emerged as central to the discussion [26]. EI is vice versa associated with distress in the form of depression, anxiety, burnout, secondary traumatic stress, emotional exhaustion and somatisation [23,27,28,29,30].

Studies have shown that individuals with a strong trait EI are more likely to engage in healthy coping behaviours when faced with stressful or other emotionally charged events [31,32]. Parents of children with T1DM have to perform many cognitive tasks related to T1DM management and at the same time deal with the intense emotions that arise from it. Trait EI and ability EI appear to predict cognitive task performance [33,34] and performance under pressure [35,36]. Studies have found a positive correlation between trait EI and objective performance/outcomes: high trait EI individuals are less implicitly prone to impulsiveness and are more self-controlled [37].

The ability to empathise with a sick child and understand the severity of the challenges posed by T1DM can have a positive effect on communication between children and parents and improve T1DM management. Empathy is one of the skills related to emotion comprehension and regulation. Empathy implies an emotional response from the comprehension of another person’s emotions and putting oneself in that person’s place, producing similar emotions to those of that person [38].

As many scientific studies show that T1DM directly affects not only the child’s but also the parents’ emotional functioning. It is very important to understand how parental EI interacts with a child’s T1DM management. We hypothesized that T1DM management may be influenced by parents’ ability to understand and control their own and others’ (most likely children’s) emotions. The aim of this study was to examine the interaction between children T1DM management and parents’ EI. We hypothesised that parents’ ability to understand and control their own and others’ emotions would be related to better paediatric T1DM management.

## 2. Materials and Methods

### 2.1. Subjects and Study Procedures

The data about parents’ EI and children T1DM management analysed below were collected in a survey conducted from 2017 to 2019 in Lithuania in the framework of the large study consisting of several stages and with multiple scientific aims. In order to form a possibly more representative research sample, we applied several sampling criteria. To be involved in the study, participants had to be mothers or primary care givers of children with T1DM. Their children had to be from 6 to 12 years old at the time of implication and had to be diagnosed with T1DM for 12 months, and had no serious comorbid medical or mental conditions. The mothers’ place of residence covered six major Lithuanian cities. Participants were selected from the Lithuanian Diabetes Registry and invited to participate in the study. Purposive sampling was used to form the pool of respondents. The final data file which includes information on 134 respondents, mothers of children aged 6 to 12 years old was formed. Thus, our study covered 134 mothers aged 25 to 56 years old. The response rate in the total sample was 89%.

Questionnaire forms for mothers were distributed and they filled the questionnaires in the outpatient department of the Lithuanian Health Science University Hospital Kaunas Clinics endocrinology department. A period of 20–30 min was provided as the time frame for filling out the questionnaires. Measures of anonymity and confidentiality were ensured. Mothers sealed the provided envelopes with the questionnaires inside after answering. Researchers were instructed about the process of carrying out the survey and how to report the number of participants.

After the completion of the questionnaire survey, the data were checked and exported to SPSS 23.0 software. The entire analysis was performed using the Complex Samples module of SPSS (version 23.0, Chicago, IL, USA), which adjusts for the complex cluster-stratified sampling method and weighted data. *p* < 0.05 was considered to be statistically significant.

### 2.2. Instrument and Variables

The research instrument consisted of questions about demographic and medical information, metabolic control EI scale and child’s medical history (e.g., T1DM history and events).

*Metabolic control.* Haemoglobin A1c (HbA1c; the main form of glycosylated haemoglobin) levels are routinely measured at clinic visits every two to three months. HbA1c is an indicator of average blood glucose concentration over the previous three months. The recommended HbA1c level for children is <7.0% [39]. The results of gHbA1c were used to evaluate the control T1DM. HbA1c shows total average of glycemia over a period of several months and therefore it is important in evaluating the effectiveness of diabetes treatment and the risk of complications. Increased blood levels of HbA1c are associated with eye, heart, kidney and nervous system disorders.

*Emotional intelligence.* In this study, we choose Trait EI model conceptualization. The EI-DARL methodology was used to assess the EI of parents of children with T1DM [40]. In our study, we used the short version of EI-DARL, which consists of 73 questions in which the degree of agreement with the statements is presented as a six-level Likert scale. Mothers had to choose the most appropriate statement. The test consists of five main scales and seven subscales.

Scales:Understanding own emotions (assesses the subject’s ability to recognize and understand one’s own emotions).Understanding others’ emotions (assesses the subject’s ability to recognize and understand other people’s emotions).Managing/regulating own emotions and behavior (assesses the subject’s ability to control own emotions).Managing/regulating others emotions and behavior (assesses the subject’s ability to control other people’s emotions).Manipulation (assesses the subject’s ability to control the behavior of others through the use of their emotions).

Subscales

Understanding the causality of own emotions (assesses the subject’s ability to understand causality of one’s own emotions).Understanding own emotions (assesses the subject’s ability to recognize and understand one’s own emotions).Transforming own negative emotions into positive ones (assesses the subject’s ability to transform one’s own negative emotions into positive ones).Self-control (assesses the subject’s ability to control one’s own emotions expression).Controlling own negative emotions (assesses the subject’s ability to control negative emotions).Understanding others’ emotions assesses the subject’s ability to understand other people’s emotions).Controlling others’ emotions (assesses the subject’s ability to control other people’s emotions).Selfish effect on others’ emotions or behavior (assesses the subject’s ability to effect other people’s emotions purposefully).Ability to evoke negative emotions in others (assesses the subject’s ability evoke negative emotions in others purposefully).

In our study, we analyzed the first four scales as a measure of the essential dimensions of EI. Manipulation scales and related subscales are not analysed, because we focused only on the positive aspects of EI, i.e., the ability to improve the situation and the ability to help others feel better.

### 2.3. Statistical Analysis

Statistical analysis was performed to impute missing data was performed to impute missing data. In our study, the amount of item-level missing data in the final sample was small (1.2%), because mean substitution performs adequately when 5% of data are missing. Means, standard deviations, bivariate correlations and internal consistency were calculated for all variables. We examined independent variables to determine their appropriateness for multivariate analyses. Two steps analysis was applied. The total sample of mothers (*N* = 134) was investigated during the first stage of the analysis in order to assess the relationship between T1DM management and EI. Cronbach’s alpha was used to evaluate the level of internal consistency reliability (Table 1) in the multi-item scale. Internal consistency was 0.92.

Logistic regression analysis was applied in order to investigate associations between diabetes management and EI. In the first step, we conducted univariable binary logistic regression (BLR) analysis. Later, cluster analysis was applied to the multidimensional analysis of the data.

### 2.4. Ethical Statement

Ethical approval for the study was provided by the Kaunas Regional Biomedical Research Ethics Committee (reference number BE-2–62). Additionally, written informed consent for participation in the questionnaire was obtained from the mothers of children with T1DM.

## 3. Results

Table 2 and Table 3 describes the demographic and socio-economic characteristics of all mothers and their children. The results show that the study sample was quite representative and gender balanced.

The data presented in Table 4 show a very weak linear statistically significant negative relationship between juvenile diabetes management and maternal EI scales. HbA1c levels decreased as parents’ total EI, ability to understand and control own emotions increased. The same tendencies were observed with the subscales of understanding the causality of own emotions, transforming negative emotions into positive ones and controlling own negative emotions. Despite statistically significant correlations, they are very weak. Therefore a deeper statistical analysis is very important.

To reveal the interactions between parental EI and T1DM management, a deeper data analysis was performed using binary logistic regression.

Binary logistic regression analysis revealed that mothers’ EI (composite index) scales of own emotions awareness and controlling own emotions were statistically reliable predictors of glycaemic control in children with T1DM. It was found that an increase in maternal EI by one point increased the probability of deteriorating glycaemic control by up to 2 times (*p* = 0.002). An increase on the scale of awareness of own emotions by one point increased the probability of deteriorating glycaemic control by up to 2.15 times (*p* = 0.04). An increase by one point on the subscale for controlling own emotions resulted in an increased probability of disease control deterioration by up to 3.1 times (*p* = 0.005), and EI subscales for understanding the causality of own emotions by up to 2.44 times (*p* = 0.018), transforming own negative emotions into positive ones by up to 2.8 times (*p* = 0.008), and controlling own negative emotions by 2.5 times (*p* = 0.013) (Table 5). Thus, an increase in the results of the EI scales and subscales listed above increases the likelihood of deterioration in T1DM management.

Meanwhile, the influence of mothers’ understanding own emotions, own controlled emotion expression, and understanding and regulation of others’ emotions regarding the child’s disease control was not statistically significant.

Cluster analysis was applied to multidimensional data analysis, using the non-hierarchical k-means cluster analysis method. Using cluster analysis, three groups of subjects with different profiles and different EI expression were distinguished. For convenience, the scales are presented on a standardized Z scale.

Figure 1 shows that the first group of respondents (24.6%) are characterized by sufficiently high values on the EI subscales. They are especially focused on their own emotional control and expression of self-control (Cluster 1/grey line). However, in comparison with other subscales, it is difficult for them to understand the causality of the own emotions and to be aware of their own emotions.

The second group of subjects (51.5%) consists of mothers whose EI values are moderate and the subscales are very similarly expressed. It is easier for them to understand their own emotions (Cluster 2/orange line).

The third group (Cluster 3/blue line) of mothers (23.9%) have generally lower EI in comparison with the other two groups. Mostly, they face challenges with their emotional control, especially transforming their own negative emotions into positive ones, but it is easier for them understand and control others’ emotions.

## 4. Discussion

This paper has focussed its scope on some of the social, family and parenting determinants of paediatric T1DM management. Therefore, the main emphasis in our analysis was made on the parents EI as the important factor determining children health indicators. As we know, parenting communication and other familial determinants can play an important role in the development of health and health behaviour in children. We aimed to investigate the interaction between paediatric T1DM management and parents’ EI, which constitutes an important component of the management of pediatric chronic disease.

Many studies have revealed the importance of EI to mental and physical health [41,42]). The present study aimed to analyse how EI interacts with disease management. Most existing research associate individuals’ levels of EI with various health indicators [43,44] in our study, we found evidence associating parents’ EI with paediatric diabetes management outcomes.

The study has established that mothers of children with sufficiently controlled T1DM have higher scores of EI scales and subscales. We found a negative statistically significant but very weak HbA1c level correlation with mothers’ ability to understand and control own emotions. The same trends were observed with the subscales of understanding the causality of own emotions, transforming negative emotions into positive ones and controlling own negative emotions. Mothers’ EI scales and subscales of understanding and regulating own’ emotions, subscales of understanding the causality of own emotions, transforming own negative emotions into positive ones and controlling own negative emotions were statistically reliable predictors of glycaemic control in children with T1DM. One of our main findings was the fact that mothers’ ability to understand and regulate own emotions, understanding the causality of own emotions and transforming own negative emotions into positive ones had significantly higher risk of complicating, not facilitating paediatric T1DM management. These results of the study suggest that higher scores on the EI scales and subscales are associated with insufficient disease control, although the opposite results were initially expected. However, the results of these studies cannot be interpreted so directly. We will discuss below several reasons may have led to such findings.

Our study shows that parents of children with insufficient T1DM management have higher EI than parents of children with sufficient T1DM. Zysberg and colleagues found that EI was associated with all glycemic management indices, i.e., haemoglobin A1c, mean blood tests per day and mean blood glucose levels [45]. Contrary to the results of our study, higher EI values in mothers resulted in better T1DM management.

In our study, a self-report test was used to evaluate mothers’ EI. Mothers had to choose the answer that suited them best on the Likert scale. Therefore, we hypothesize that the results of the EI study may more accurately reflect mothers’ subjective attitudes toward their ability to know and control their own and others’ emotions. In addition, the issues in this methodology are not child-specific. Due to the peculiarities of the child’s psychological development, adult-friendly strategies may not be suitable for communication with children at all. The EI self-report test results may reflect mothers’ efforts, but not their abilities or skills to know, understand and control own emotions. The tendency of women to identify their maternal self-esteem with the results of child’s DTM1 management may have influenced the results of this study: mothers with insufficient T1DM management may have felt guilty and tried to show themselves more positively. A sense of guilt can have an annoying thinking effect, i.e., constantly analyzing difficulties and how these affect emotions. It is important to note that high levels of HbA1c can negatively affect a child’s emotional balance and behaviour. Therefore, mothers really need to struggle with more every day parenting challenges, requiring greater understanding and control of own emotions. The results of the study about mothers’ ability to control own negative emotions agrees with positive effect of mothers’ EI on the control of a child’s diabetes management [46].

We hypothesized that the association of mothers’ EI’s scales and subscales of understanding and managing own emotions with T1DM management would be associated with emotional coping resources. We repeat the hypothesis put forward by Zysberg and his colleagues, in that parents with higher EI levels can better identify their child’s emotional cues, relate and communicate more effectively, and manage the challenges of daily monitoring and care administration with greater ease [45]. Such cognitive coping strategies [47], i.e., positive refocusing (i.e., shifting one’s thoughts to focus on something positive), refocusing on planning (i.e., engaging in problem-solving), acceptance (i.e., accepting that the event happened) and positive reappraisal (i.e., attaching a positive meaning to the event) may relate to the ability to regulate one’s own and others’ emotions.

The results of our study suggest that mothers’ ability to understand the emotions of others may complicate the management of a child’s illness. Many studies show the importance of parental empathy in managing a child’s illness, while understanding that emotions are a component of empathy [48]. Cognitive strategies like putting events into perspective (i.e., thinking of the event in the context of other, more difficult events) related to empathy and the EI dimension to the ability to understand the emotions of others. Thus, helping their child to cope with distress over treatment may increase adherence to treatment and improve management outcomes. In emotional background thoughts and reflections used in dialogue with children, there may be strategies for dealing with their emotional strain, but these strategies should be adopted to paediatric psychosocial maturation and associated with parenting skills. These insights may partly explain the results of our study.

In our study, the research outcomes did not allow us to objectively identify what coping strategies mothers used to control own emotions. Self-report tests reflect subjective behavioural evaluations and EI questions are oriented toward everyday situations with adults. The EI dimension about the ability to control own emotions may be a coping resource. We hypothesise that the EI scale of own emotions controlling may be closely related to cognitive regulation of emotions. The current literature on EI and coping suggests that the concept of emotional regulation may be considered as a coping [49]. Cognition plays an important role in the management of one’s own and others’ emotions, because cognitions are conscious cognitive strategies to think about one’s emotional reactions and how they influence possible future coping behaviour [47]. In further research, it will be important to look for interactions between EI and coping strategies. Mothers may try to understand and control their emotions, but coping strategies may not be appropriate for the challenges of diabetes.

The results of the present study may also reflect an inefficient distribution of responsibility for disease management between the child and the mother. A biomedical treatment model still prevails, but this underestimates the psychological aspects of the disease, as a result of a lack of health care professionals involved in health care teams [50]. As a result, the child’s own active participation in the treatment process is greatly underestimated [51] and too much responsibility is transferred to the parents, especially mothers. Therefore, mothers may be inclined to blame themselves, while it is important to enable the child to take over the management of the disease consistently and gradually. It is very important for this process to deepen how the child feels, and to help the child manage emotions, stress coping and problem-solving strategies [52].

### Strengths and Limitations

The strength of this study is being one of the first attempts to understand the influence of parents’ EI on the management of a child’s disease. The first article on this type of study appeared in 2013 [45]. Other subsequent studies about the influence of EI on diabetes management have been performed with patients who were mostly adults and in most cases had type 2 diabetes [53,54]. The novelty of this study is that the role of EI in the management of childhood disease has been based on theories of trait EI, whereas Zysberg and colleagues relied on the view of EI as an ability.

The present study has a few limitations requiring consideration when interpreting the results. Only the HbA1 test result was used to assess disease management. There has been a lot of discussion lately about the objectivity of this test in assessing disease control [54,55,56]. HbA1 shows only a three- to four-month mean glycaemic index, not glycaemic fluctuations. Optimal glycaemic rates should be constant; the mean may mask persistent glycaemic spikes, which are signs of poor disease management. Frequent and consistent glycaemic control is essential for good quality disease management. It is appropriate to include daily glycaemic averages and glycaemic measurement frequencies as disease management indicators.

Other limitation of this study may be using of EI self-report measures instead of including performance-based measures. Self-report may be a weaker indicator than a performance-based test. For example, Brackett and Mayer (2003) show that self-report measures and performance-based tests show little to no association and display differential associations with various criteria [57]. In a study by Zysberg and colleagues [45], where self-report measures and performance-based measures were used, the performance-based measures were related to all disease management indicators, while self-reported measures revealed a link between EI and only one disease management indicator [45]. Our observations and insights coincide with those of Zysberg and colleagues: it is very important to consider that self-report EI will mainly associate with own self-report outcomes, while performance measures may associate with less subjective outcomes.

The EI-DARL methodology is Lithuanian and has not been used in foreign or international studies before, so we cannot compare the results of this study with other foreign studies. This study is an attempt to better understand what internal resources of parents are helpful for coping with the day-to-day challenges of T1DM.

The results of this study can be used scientifically and practically. Further EI re-search can be conducted using performance-based assessment and self-report measures. It would also be useful to compare parents’ EI indicators with their own and children’s emotional health indicators: depression, anxiety, fear. These comparative studies would provide a better understanding of EI significance.

The practical significance of this study may be the attention of health professionals to the adjustment of parental EI to child’s T1DM management. It is worth noting that parents deal with huge amount of their own, sick child’s and others family members’ emotional load on a daily basis. Efforts to understand and control one’s own and others’ emotions are not enough. It is very important to teach parents specific strategies to overcome the emotional burden or adapted to life with a child’s T1DM. Parenting skills training programs, which include the teaching and development of emotional literacy related to the child’s emotional needs, can be a great help in this.

In further research, it is appropriate to delve into the parents’ specific strategies to understand and control emotions in interaction with their children with T1DM; how EI interacts with indicators such as depression, anxiety, and fear of parents and their sick children; how EI interacts with psychological well-being and parenting skills, parenting style; how parental EI relates to a child’s disease management skills, subjectively perceived competence to take care of their own health; how parental EI interacts with parents’ and children’ s stress and problems related to T1DM management coping strategies; how programs such as parenting skills development programs can operate on parental EI.

## 5. Conclusions

Family environments and parents’ ability to deal with their own emotional strain and that of their children constitute a foundation for the further development of a healthy lifestyle during childhood. These traits are especially important when confronted with a paediatric chronic illness.

Our study showed that the EI of mothers plays a very important role in paediatric T1DM management. The results of the study cannot be explained easily also because of the complex nature of T1DM, its management and effect on parent-child relationships. With some exceptions, the majority of our findings are in contrast to our hypothesis and the findings of other researchers. The results of this study suggests mothers’ EI scales and subscales of understanding and regulating own emotions, subscales of transforming own negative emotions into positive ones and controlling own negative emotions are statistically reliable predictors of glycaemic control in children with T1DM: it was found that an increases in maternal EI, the scale of awareness of own emotions, the subscales for controlling own emotions, understanding the causality of own emotions, transforming own negative emotions into positive ones and controlling own negative emotions increased the probability of deteriorating glycemic control. The results of our study demonstrate that mothers’ EI is a critical component in paediatric T1DM management.

## Figures and Tables

**Figure 1 ijerph-18-03117-f001:**
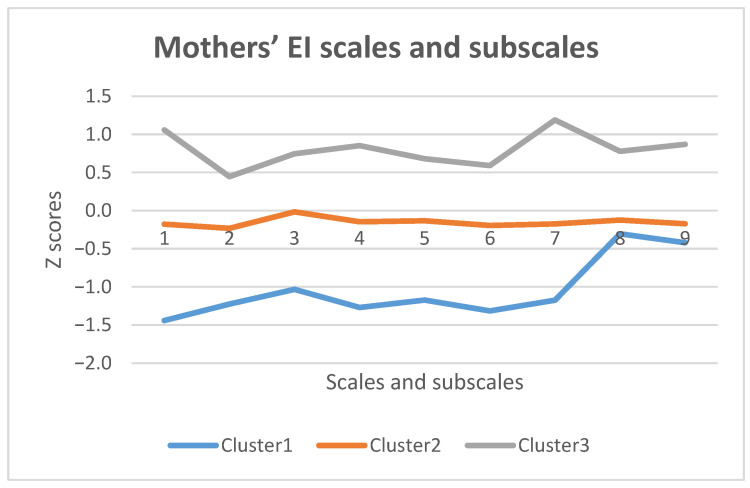
Mothers’ EI scales and subscales cluster analysis. 1. Control of own emotions; 2. Understanding the causality of own emotions; 3. Understanding own emotions; 4. Transforming own negative emotions into positive ones; 5. Controlling own negative emotions; 6. Own emotion awareness; 7. Self-control expression; 8. Understanding others’ emotions; 9. Controlling others’ emotions.

**Table 1 ijerph-18-03117-t001:** The level of internal consistency reliability of emotional intelligence (EI) scales and subscales.

EI Scales and Subscales	Cronbach’s Alpha
Total value of emotional intelligence	0.92
A scale of awareness of own emotions	0.89
A scale of controlling own emotions	0.90
A scale of awareness of others’ emotions	0.84
A scale of controlling others’ emotions	0.81
Understanding the causality of own emotions	0.88
Understanding own emotions	0.89
Transforming own negative emotions into positive ones	0.78
Self-control	0.82
Controlling own negative emotions	0.76
Understanding others’ emotions	0.84
Controlling others’ emotions	0.81

**Table 2 ijerph-18-03117-t002:** Demographic and social characteristics of the total sample and subsample studied.

Independent Variables	*N*	%
Total sample studied, *N* = 134		
Child’s sex		
Girls	69	51.5
Boys	65	48.5
Marital status		
Married	114	85.1
Divorced	12	9.0
Live separately	1	0.7
Widow	3	2.2
Single mother	4	3.0
Parental education		
Basic/less than high school	3	2.2
Secondary/high school	15	11.2
Special secondary	5	3.7
College/diploma	30	22.4
University/graduate degree	81	60.4
Financial situation of the family		
Very good	7	5.2
Good	91	67.9
Moderate	33	24.6
Poor	3	2.2

**Table 3 ijerph-18-03117-t003:** Demographic characteristics of the total sample and subsample studied.

Independent Variables	Mean	Median	Variance
Total sample studied, *N* = 134			
Parents age	37.83	37.00	19.13
Number of children in family	1.98	2.00	0.68
Children age	9.26	9.00	4.16
Duration of sickness	3.53	3.00	5.92
Glycemic control	7.43	7.3	1.91

**Table 4 ijerph-18-03117-t004:** Correlation between children type I diabetes (T1DM) management and EI scales and subscales of mothers (*N* = 134).

Independent Variables	Correlation	*p*<
Total value of emotional intelligence	−0.19	0.02 *
A scale of awareness of own emotions	−0.15	0.03 *
A scale of controlling own emotions	−0.15	0.03 *
A scale of awareness of others’ emotions	−0.06	0.38
A scale of controlling others’ emotions	−0.07	0.3
A subscale of understanding the causality of own emotions	−0.17	0.01 *
A subscale of understanding own emotions	−0.02	0.71
A subscale of transforming own negative emotions into positive ones	−0.15	0.03 *
A subscale of own controlled emotion expression	−0.10	0.14

Spearman linear correlation coefficient. * Correlation is significant at the 0.05 level (two-tailed).

**Table 5 ijerph-18-03117-t005:** Correlation between children T1DM management and EI scales and subscales of mothers (*N* = 134).

Predictors	Sufficient Diabetes Management	Insufficient Diabetes Management	Univariable Logistic Regression	
Diabetes management	*n* (%)	*n* (%)	OR	CI
	72 (53.7)	62 (46.3)		
Total value of emotional intelligence	32 (74.4%)	40 (44.0%)	**1**	**0.307–0.76**
	11 (25.6%)	51 (56.0%)	**0.48**	
A scale of awareness of own emotions	27 (62.8%)	40 (44.0%)	1	0.22–0.97
	16 (37.2%)	51 (56.0%)	**0.46**	
A scale of controlling own emotions	44 (48.4%)	32 (74.4%)	1	**0.14–0.71**
	47 (51.6%)	11 (25.6%)	**0.32**	
A scale of awareness of others’ emotions	38 (88.4%)	69 (75.8%)	1	0.14–1.178
	5 (11.6%)	22 (24.2%)	0.41	
A scale of controlling others’ emotions	21 (48.8%)	30 (33.0%)	1	0.24–1.08
	22 (51.2%)	61 (67.0%)	0.51	
Understanding the causality of own emotions	26 (60.5%)	35 (38.5%)	1	**0.19–0.85**
	17 (39.5%)	56 (61.5%)	**0.4**	
Understanding own emotions	43 (100%)	84 (92.3%)	1	0.000-
	0 (0%)	7 (7.7%)	8.27 × 10^8^	
Transforming own negative emotions into positive ones	31 (72.1%)	43 (47.3%)	**1**	**0.15–0.75**
	12 (27.9%)	48 (52.7%)	**0.35**	
Own controlled emotion expression	27 (62.8%)	41 (45.1%)	1	0.23–1.02
	16 (37.2%)	50 (54.9%)	0.48	
Controlling own negative emotions	26 (60.5%)	34 (37.4%)	**1**	**0.18–0.82**
	17 (39.5%)	57 (62.6%)	**0.39**	

Significant relationships are provided in bold. OR—odds ratio; 95% CI—95% confidence interval.

## Data Availability

Not applicant.

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
