# Peer review of "Paediatric Type1 Diabetes Management and Mothers’ Emotional Intelligence Interactions"

_ijerph, 2021, doi:10.3390/ijerph18063117_

Round 1
Reviewer 1 Report
This article proposes pediatric type1 diabetes management and mothers’ emotional intelligence interactions. The test consists of five main scales and seven subscales. However, these test conditions are unclear. The authors should clearly define these tests. In Fig. 2, what mean for Y-axis?
And then, the following items should be explained in detail.
1. Control of own emotions
2. Understanding the causality of own emotions
3. Understanding own emotions
4. Transforming own negative emotions into positive ones
5. Controlling own negative emotions
6. Own emotion awareness
7. Self-control expression
8. Understanding others’ emotions
9. Controlling others’ emotions
Moreover, relationships should be dicussed.
Author Response
Dear Madam or Sir,
Please see the attachment.
Sincerely,
Jolanta Zilinskiene

Reviewer 2 Report
I would appreciate if the conclusions and results of the investigation could be drafted and detailed more clearly. It seems to me a very original and decisive investigation. However, I raise an important question: What is the use of the data obtained? Also, how and where can they be applied? What is the reason for this investigation? What are the justification and its application?
On the other hand, what future lines of research are suggested?
Author Response

(The authors gave the same response as above.)

Round 2
Reviewer 1 Report
The authors have addressed all of my concerns. Hence, I think that the revised article can be accepted for publication in this journal.